# Development of deep learning-based detecting systems for pathologic myopia using retinal fundus images

Li Lu [1,7], Enliang Zhou[2,7], Wangshu Yu[1,7], Bin Chen[3], Peifang Ren[1], Qianyi Lu[4], Dian Qin[5], Lixian Lu[5], Qin He[1], Xuyuan Tang[1], Miaomiao Zhu[1], Li Wang[6] & Wei Han [1✉]

Globally, cases of myopia have reached epidemic levels. High myopia and pathological myopia (PM) are the leading cause of visual impairment and blindness in China, demanding a large volume of myopia screening tasks to control the rapid growing myopic prevalence. It is desirable to develop the automatically intelligent system to facilitate these time- and labor-consuming tasks. In this study, we designed a series of deep learning systems to detect PM and myopic macular lesions according to a recent international photographic classification system (META-PM) classification based on color fundus images. Notably, our systems recorded robust performance both in the test and external validation dataset. The performance was comparable to the general ophthalmologist and retinal specialist. With the extensive adoption of this technology, effective mass screening for myopic population will become feasible on a national scale.

[1] Department of Ophthalmology, The First Affiliated Hospital, School of Medicine, Zhejiang University, Hangzhou, Zhejiang, China. [2] Department of Ophthalmology, The First Affiliated Hospital of University of Science and Technology of China, Hefei, Anhui, China. [3] Department of Ophthalmology, Changshu First People's Hospital Affiliated to Soochow University, Changshu, Jiangsu, China. [4] Department of Ophthalmology, The First Affiliated Hospital of Soochow University, Suzhou, Jiangsu, China. [5] College of Computer Science and Technology, Zhejiang University, Hangzhou, Zhejiang, China. [6] Hangzhou Zhicheng Technology Inc, Hangzhou, Zhejiang, China. [7] These authors contributed equally: Li Lu, Enliang Zhou, Wangshu Yu. ✉email: hanweidr@zju.edu.cn

Myopia is a worldwide public health problem. It was estimated that nearly 2.5 billion people might be affected by 2020, among which 27–33% would be high myopia[1]. By 2050, the global myopic population can reach 4.8 billion (49.8% of the global population)[2], leading to the substantial global economic loss due to the burden of visual impairment caused by myopia[3].

Pathologic myopia (PM) is one of the most common causes of severe irreversible visual impairment due to the myopic maculopathy (MM). According to global population surveys, the incidence of PM was 5–10% in diagnosed myopia and 1–4% in the general population[4–6]. In particular, the situation in Eastern Asia nations is worrisome. In China, more than 50% students in urban have myopia and 20% are high myopia[4,7]. Although the typical PM lesions can also be observed in some moderate myopia[8], high myopia was generally thought as the essential condition for PM and people with high myopia are at higher risk of PM[9].

As the treatment effects and prognosis of PM are often poor, early detection and intervention to prevent the aggravation of PM are critically important. However, the retinal specialists are always confronted with heavy clinical duty managing multiple ocular fundus diseases, while the healthcare institution workers are often lack of the requisite expertise to identify the PM and MM, let alone the management. Moreover, currently some governments had launched the public healthcare programs to screen the ocular fundus diseases including PM in local community medical institutions[10–12]. Such large screening task further challenges the ophthalmic medical resources with respect of clinical data analysis, especially the retinal fundus images reading. The work volume of image data processing will be daunting if done by man power alone.

Fortunately, with the rapid development of artificial intelligence (AI) technologies, the application of AI in medical areas started to play important role in the automated clinical data processing and hence made the heavy work feasible[13]. Among AI technologies, the deep learning, a sophisticated subclass of machine learning, mimics the way of human brain working and uses deep artificial neural networks to solve any feature expression problem[14]. The deep learning system (DLS) has shown excellent performance comparable to the board-certified specialists in the respect of massive medical images categorizing in clinical practice[15–17]. In the field of ophthalmology, the DLS-affiliated diagnosis software for diabetic retinopathy has been successfully applied in clinical and public healthcare screening task[12].

However, the application of deep learning technology in PM lesions screening is still a challenge, due to the complexity of classification and definition system of PM. For a long time, there had been a lack of consensus on the precise definition of PM, until a new simplified systematic classification for MM was proposed by a meta-analysis for pathologic myopia (META-PM) study group[18]. According to this classification standard, eyes with MM, which is equal to or more serious than diffuse choroidal atrophy (category 2) or with at least one of the "plus" lesions, can be defined as having PM[19]. Based on such classification system, it is viable to design the AI algorithm to automatically identify PM and assist the clinicians to make relevant diagnosis. Importantly, sufficient resources of high-quality PM retinal fundus image dataset and high standard expert teams are also crucial to achieve this goal.

This study aimed to design and train the DLSs to automatically detect PM as well as the category of MM using a large dataset of color retinal fundus images obtained from the ophthalmic clinics of the hospitals.

## Results

Altogether 16,428 gradable images were screened from 17,330 color fundus images and used for PM grading, while 902 images were labeled as ungradable. The workflow of manual grading tasks was illustrated in Fig. 1. About 15% of the graded images with inconsistent diagnosis were submitted to retinal specialists for final grading. The characteristics and summary of all the datasets were shown in Table 1 and Supplementary Table 1.

**Performance of the DLSs.** In the test dataset for binary task (none PM /PM), the external general ophthalmologist had an accuracy of 0.978 (95% CI: 0.971–0.985), sensitivity of 0.980 (95% CI: 0.973–0.986), and specificity of 0.967 (95% CI: 0.958–0.976), while the external retinal specialist achieved an accuracy of 0.991 (95% CI: 0.986–0.996), sensitivity of 0.991 (95% CI: 0.987–0.996), and specificity of 0.989 (95% CI: 0.984–0.994). The DLS achieved an AUC of 0.993 (95% CI: 0.989–0.997), accuracy of 0.977 (95% CI: 0.970–0.984), specificity of 0.972 (95% CI: 0.962–0.980), and sensitivity of 0.977 (95% CI: 0.970–0.985) (Table 2, Fig. 2a).

In the test dataset for three-class classification task (ungradable/NPM/PM), the DLS achieved a performance measured with the macro-AUC value of 0.979, an accuracy of 0.963, and quadratic-weighted kappa of 0.787 (Fig. 2b). Meanwhile, in the test dataset for five-class task, the DLS achieved a macro-AUC of 0.978, accuracy of 0.976, and quadratic-weighted kappa of 0.990 (Fig. 2c). From C0 to C4 MM, the specific accuracy of DLS is 98.8, 99.3, 93.7, 95.5, and 93.9% respectively. Overall, the DLS achieved the performance which was comparable to the general ophthalmologist. The detailed results of performance in multiclass classification tasks were listed in Table 3. Moreover, the performance of DLSs in the external validation dataset was slightly inferior to that in the test dataset (Supplementary Table 2). The precision recall curves of DLSs were also provided in Supplementary Fig. 1 (the source data were shown in Supplementary Data 1).

**Misclassified images analysis.** In the binary task of NPM/PM, the total number of false-negative classification was five, among which two were PM with retinal detachment and three were PM with retinal vein obstruction. Of the thirty-three false-positive images, thirty were tessellated fundus characterized as PM, while the remaining three were retinal vein obstruction, proliferative retinopathy, and exudative retinopathy. In the multiclass classification tasks, the results were shown in confusion matrices (Supplementary Fig. 2). The major error in the three-class task was that fifty-five NPM images were misclassified as ungradable by the DLS due to the poor clarity. Meanwhile, in the five-class task, the major error was that two patchy chorioretinal atrophy images were erroneously classified as macular atrophy. Similarly, there were two images of macular atrophy that were incorrectly identified as patchy chorioretinal atrophy. The typical misclassified images were shown in Fig. 3. In addition, the confusion matrices of three tasks in the external dataset were also provided in Supplementary Fig. 2. The distribution of misclassified images for the 3 tasks in the external dataset was basically comparable to that in the test dataset.

**Visualization analysis.** Heatmap analysis demonstrated the learning procedure of our DLS and the areas contributing to the DLS. The original images of different MM categories in the five-class task were displayed in Fig. 4a. After overlying fundus heatmap in Fig. 4a, the regions that the DLS thought most significant for its decision were highlighted and displayed in Fig. 4b. The typical MM lesions including tessellated fundus, diffuse

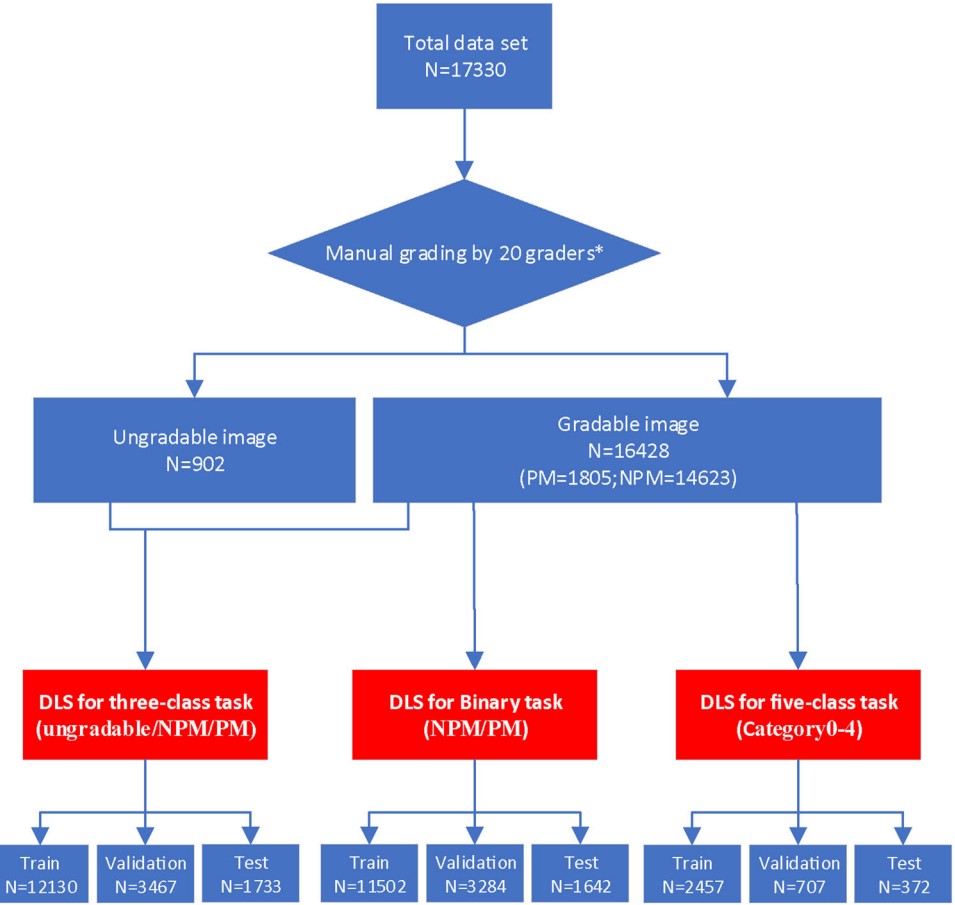

**Fig. 1 Workflow diagram showing the overview of developing deep learning systems to detect PM as well as myopic maculopathy.** PM pathologic myopia, NPM none pathologic myopia, DLS deep learning system. *20 graders were randomly grouped into five teams with each team involving three general ophthalmologists and one senior specialist.

**Table 1 Study population characteristics of the total dataset and external validation dataset.**

|  | Number of images with labels | Number of participants | Mean age (years) | Sex (% female) | Spherical equivalent (diopters) |
|---|---|---|---|---|---|
| Total dataset | 17,330 | 13,869 | 49.5 | 66.1 | −2.6 ± 4.79[a] |
| Ungradable images | 902 | 881 | 48.8 | 66.6 | NA |
| none PM | 14,623 | 11,698 | 49.2 | 65.9 | −1.294 ± 2.39 |
| Pathologic myopia | 1805 | 1290 | 52.7 | 67.5 | −14.469 ± 4.84 |
| Category 0 | 693 | 645 | 50.6 | 61.2 | −7.23 ± 0.22 |
| Category 1 | 1581 | 1089 | 48.9 | 66.3 | −11.38 ± 2.75 |
| Category 2 | 480 | 338 | 49.7 | 67.5 | −14.06 ± 4.26 |
| Category 3 | 451 | 334 | 55.9 | 68.9 | −16.26 ± 5.24 |
| Category 4 | 331 | 188 | 61.3 | 67 | −16.87 ± 5.78 |
| External validation dataset | 1000 | 738 | 51.5 | 63.4 | −3.07 ± 5.80[a] |
| Ungradable images | 63 | 59 | 59.0 | 63.9 | NA |
| none PM | 800 | 602 | 50.5 | 64.2 | −1.5 ± 3.42 |
| Pathologic myopia | 137 | 77 | 53.6 | 55.8 | −15.35 ± 5.98 |
| Category 0 | 35 | 31 | 52.1 | 64.1 | −6.73 ± 0.24 |
| Category 1 | 121 | 78 | 49.8 | 61.6 | −12.35 ± 4.98 |
| Category 2 | 32 | 17 | 50.5 | 62.7 | −15.75 ± 5.71 |
| Category 3 | 33 | 18 | 56.2 | 65.1 | −17.03 ± 5.96 |
| Category 4 | 23 | 13 | 62.4 | 66.2 | −17.26 ± 6.21 |

Abbreviations: *PM* pathologic myopia.
[a]not include the refractive error data of ungradable image group.

**Table 2 Classification results for binary task in test dataset.**

|  | AUC (95% CI) | Accuracy (95% CI) | Specificity (95% CI) | Sensitivity (95% CI) |
|---|---|---|---|---|
| DLS | 0.993 (0.989 to 0.997) | 97.7% (97.0 to 98.4) | 97.2% (96.2 to 98.0) | 97.7% (97.0 to 98.5) |
| General ophthalmologist[a] | – | 97.8% (97.1 to 98.5) | 96.7% (95.8 to 97.6) | 98.0% (97.3 to 98.6) |
| Retinal specialist[a] | – | 99.1% (98.6 to 99.6) | 98.9% (98.4 to 99.4) | 99.1% (98.7 to 99.6) |

Abbreviations: *DLS* deep learning system, *AUC* area under the receiver operating curve.
[a]The external ophthalmologist and retinal specialist.

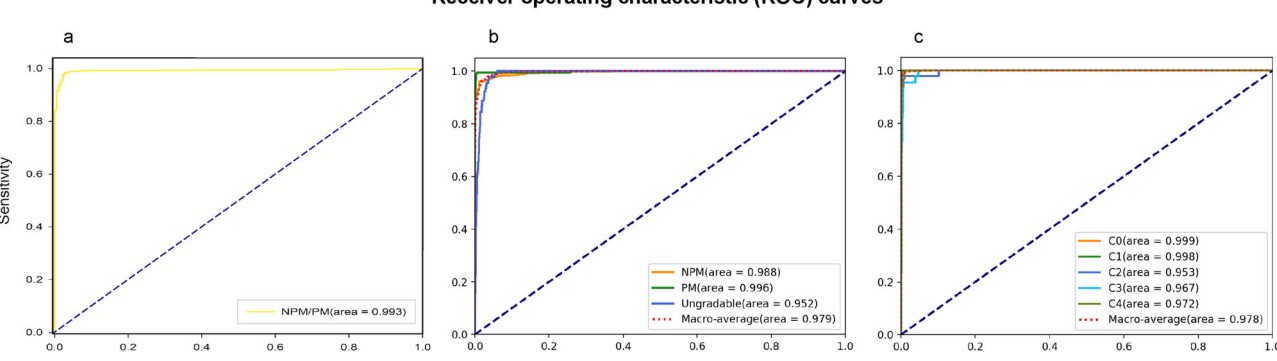

**Fig. 2 Receiver operating characteristic (ROC) curves of the deep learning systems derived from the test datasets. a** The performance for binary task. **b** The performance for three-class task. **c** The performance for five-class task. NPM none pathologic myopia, PM pathologic myopia, AUC area under the receiver operating curve, C Category.

chorioretinal atrophy, patchy chorioretinal atrophy, and macular atrophy were observed in the hot regions. Of note, these lesions were used to diagnose PM in ophthalmic practice. The t-SNE analysis showed the excellent classification performance of our DLSs for each task. In the t-SNE plots, clustering occurred in the same classification of images, with significant differences between images with different labels (Supplementary Fig. 3).

## Discussion

In view of the rapid increase in myopia prevalence[7,20,21], there was a demand to establish the systemic electronic ocular databases especially for the adolescent individuals. Meanwhile, the large amount of PM patients also should receive the timely diagnosis and advices by eye doctors to avoid the occurrence or aggravation of ocular complications. In particular, with the serious situation of COVID-19 pandemic, the remote medical systems will be more broadly applied. To implement such high throughput or telemedicine task, the aid of modern AI technologies is crucial.

In this study, we designed a series of DLSs incorporating three tasks to detect PM and the category of MM based on retinal fundus images. Our work was an exploratory and innovative effort to apply the modern deep learning technologies in the PM diagnosis and management. The target serving population for clinical application of our DLS will mainly be the subjects with myopia, as the DLS is designed to identify the PM. Using a large dataset of 17,330 retinal fundus images labeled by ophthalmologists and senior retinal specialists from real-world, our DLSs achieved robust performance in the binary task of NPM/PM (AUC = 0.993), the three-class task of ungradable/NPM/PM (macro-AUC = 0.979) and the multiclass classification task of five MM categories (macro-AUC = 0.978). Besides, based on quadratic-weighted kappa value, the DLS showed slightly less consistent with the reference standard than the general ophthalmologist in the three-class task and demonstrated consistency equivalent to that of a retinal specialist in the five-class task. The competitive performance of our DLSs in this work could mainly be

attributed to two factors: the suitable CNN architecture and larger training dataset. We selected Xception inspired by InceptionV3 as the basic architecture after validation. In this architecture, Inception modules were replaced with depthwise separable convolutions to handle spatial-correlations and cross-channel correlations independently. It is believed that an independent processing of corss-channel correlations would better address the tasks involved in this study. Due to the strong pixel similarity in the characteristic PM lesions, Xception can better learn the characteristic similarity of multiple pixel points within a PM lesion to make more accurate judgments by processing cross-channel correlations independently. Although the number of parameters was similar to InceptionV3, Xception can take better use of model parameters[22]. On the other hand, the larger training dataset was used in this work and ought to be effective to prevent parameters overfitting. In the process of DLS development, in view of the relatively small number of training images for some categories, several strategies were adopted to avoid overfitting. Firstly, the diversity of the training data were increased by means of data augmentation by randomly flipping training images from left to right or up to down, and adjusting images' brightness, contrast, hue, and saturation. Secondly, dropout layer were inserted between the basic Xception model and the last full connection layer. The dropout ration is 0.2 with only 20% of the weights updated every time. Thirdly, we used a two-step training process. In the first training step, only the last full connection layer was fine-tuned and the weights of the basic Xception were not updated. In the second training step, the whole model was fine-tuned and the weights of all the layers were updated. Finally, we used the early stopping strategy. Training was stopped if the validation loss did not decrease in 5 consecutive epochs during the training process. Therefore, our DLSs were proven to be a precise and efficient solution for the PM screening task.

AI and big data analysis have currently been widely applied in the medical field. Deep learning technologies based on image data have become the powerful tools in screening diabetic retinopathy (DR), glaucoma, retinopathy of prematurity (ROP), and age-related macular degeneration (AMD)[23–26].

**Table 3 Classification results for multiclass tasks in test dataset.**

|  | Macro-AUC | Accuracy(95% CI) | Quadratic-weighted kappa(95% CI) |
|---|---|---|---|
| Task of ungradable/NPM/PM | | | |
| DLS | 0.979 | 96.3% (95.1 to 97.5) | 0.787 (0.737 to 0.837) |
| General ophthalmologist | – | 98.4% (97.6 to 99.2) | 0.962 (0.940 to 0.979) |
| Retinal specialist | – | 99.2% (98.6 to 99.8) | 0.981 (0.969 to 0.994) |
| Task of 5 myopic maculopathy categories | | | |
| DLS | 0.978 | 97.6% (96.8 to 98.3) | 0.990 (0.985 to 0.994) |
| Category 0 | – | 98.8% | – |
| Category 1 | – | 99.3% | – |
| Category 2 | – | 93.7% | – |
| Category 3 | – | 95.5% | – |
| Category 4 | – | 93.9% | – |
| General ophthalmologist | – | 95.4% (94.3 to 96.4) | 0.966 (0.957 to 0.974) |
| Category 0 | – | 97.7% | – |
| Category 1 | – | 98.1% | – |
| Category 2 | – | 91.6% | – |
| Category 3 | – | 88.8% | – |
| Category 4 | – | 90.9% | – |
| Retinal specialist | – | 98.9% (98.3 to 99.4) | 0.991 (0.986 to 0.995) |
| Category 0 | – | 100% | – |
| Category 1 | – | 99.3% | – |
| Category 2 | – | 97.9% | – |
| Category 3 | – | 97.7% | – |
| Category 4 | – | 96.9% | – |

Abbreviations: *DLS* deep learning system, *PM* pathologic myopia, *NPM* none pathologic myopia.

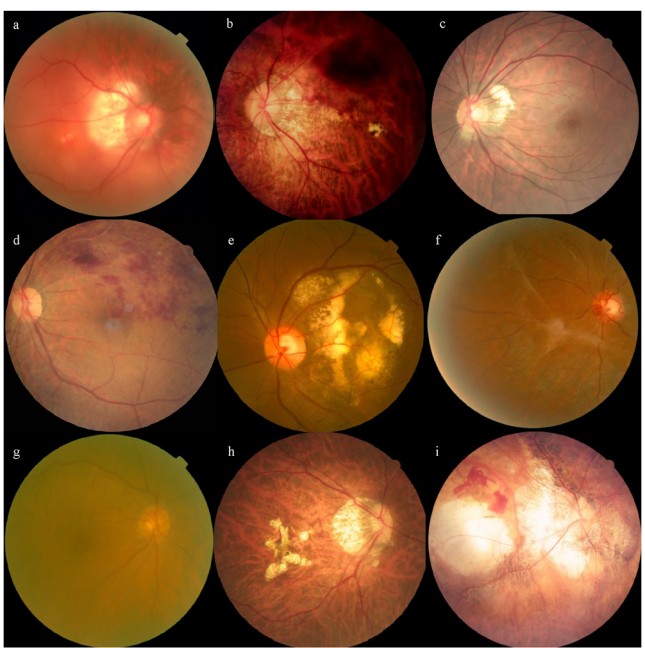

**Fig. 3 The typical misclassified cases of DLSs.** Typical images of false-negative images in binary task: **a** PM with retinal detachment. **b** PM with retinal vein obstruction. Typical images of false-positive images in binary task: **c** Tessellated fundus. **d** Retinal vein occlusion. **e** Exudative retinopathy. **f** Proliferative retinopathy. The major error cases of three-class task: **g** images with the relatively poor clarity. The major error cases of five-class task: **h** patchy chorioretinal atrophy image were classified as macular atrophy image. **i** macular atrophy image were classified as patchy chorioretinal atrophy image.

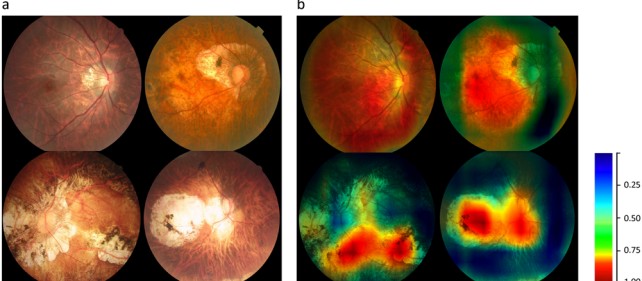

**Fig. 4 Visualization of the DLS for five-class task. a** The original images of different myopic maculopathy (Category1–Category4). **b** Heatmap generated from deep features overlaid on the original images. The typical myopic maculopathy lesions were observed in the hot regions.

Several automated systems for detection of PM have been reported. For instance, Tan et al. introduced the PAMELA system that automatically detected PM via peri-papillary atrophy feature[27]. Zhang et al. proposed a computer-aided diagnosis framework for PM diagnosis through biomedical and image informatics[28]. Freire et al. reported the work of PM diagnosis and detection of retinal structures and some lesions using Xception as the baseline model and achieved satisfactory performance in classification and segmentation tasks[29]. Devda et al. developed a deep learning method with CNN for tasks of Pathologic Myopia Challenge (PALM) based on dataset provided by International Symposium on Biomedical Imaging (ISBI). Their works showed a better performance when compared to PAMELA system[30]. However, these were all developed from public databases with small sample sizes, such as Singapore Malay Eye Study (SiMES) database, Singapore Cohort Study of the Risk factors for Myopia (SCORM) and ISBI. The datasets in this study were collected from the real-world of eye center, which provide more original disease information and data complexity compared with the public databases. Nevertheless, in real-world screening conditions, the rate of detecting ungradable images or poor-quality images exist, being reported at a level about 20%[31,32]. In the present work, using targeted three-class task training, the DLSs recorded high performance in distinguishing ungradable images or poor-quality images from the raw total dataset (Fig. 2b, Table 3).

In clinical practice, the category of MM can reflect the severity of the disease to a large extent. Zhao et al. reported that the morphological and functional characteristics of highly myopic eyes were positively correlated to the MM severity from Category 0 to 3. Meanwhile, the complications were different from Category 0 to 4 and closely correlated to the MM degree[33].

A recent study by Kyoko et al. developed four DL models to recognize the Category 2 to 4 MM and CNV. They also integrated the four DL models and a special processing layer to develop the META-PM categorizing system (META-PM CS), which could detect PM defined as having MM equal to or more serious than diffuse atrophy (category 2)[34]. Compared with their META-PM CS, our five-class DLS is able to recognize MM from Category 0 to 4 directly. The binary task of NPM/PM was assigned to a well-trained DLS based on a more precise PM definition (equal to or more serious than diffuse atrophy (category 2) or with at least one of the "plus" lesions). Therefore, our DLSs had prominent significance for ocular healthcare work. The category of MM lesions suggested by our DLS in the five-class task can provide the automatic medical diagnosis and relevant

medical advice for patients instantly. Meanwhile, the DLSs also can assist ophthalmologists to obtain the prompt information of MM lesions for reference in their clinical practice.

The interpretation of deep learning remains a long-standing problem[35]. This study analyzed the distribution of misclassification of the DLS, including false negatives and false positives, and visualized the output images (Figs. 3 and 4). The misclassified images in our results indicated the weakness of DLSs somewhat. However, the misclassification rate of our three DLSs was within 4%. The outcomes at this stage were optimist for future's clinical application. Certainly, increasing the image number of specific diseases into the training dataset will always be the effective approach to further minimizing the false-positive and false-negative results. Moreover, the visualization results in heatmap indicated the regional contribution to decision making by DLS on an image. The typical MM lesions found in the hot regions of the heatmap (Fig. 4) indicated the central procedure of the DLSs and confirmed the validity of our systems. The results also suggested the direction of optimization and updating for the better precision in the future.

Although the DLSs designed by our work demonstrated excellent performance, this study still had limitations. Firstly, our DLSs had no independent function to detect "plus" lesions including CNV, lacquer cracks, and Fuchs spots. These three specific lesions differ greatly in shape, size, color, and location, especially the lacquer cracks. A large number of images containing these "plus" lesions are required to train the DLS in a segmentation task, which is another ongoing work including images collection and labels annotation by our team. Secondly, we had a single source of the total dataset in this study. Although we tested our DLSs in the external validation dataset and achieved robust performance, massive external validation was still desirable to ensure its successful application in the real world of PM screening in the future. Thirdly, the DLSs were exclusive one for PM identification and not trained to identify the ocular diseases other than PM. It was not an automated comprehensive diagnostic platform to screen multiple fundus diseases.

In conclusion, this study designed the DLSs which can automatically detect the PM and the category of MM according to the META-PM study classification. The data of our work at this stage were promising. The DLSs achieved excellent performance in the test and external validation datasets. This pilot work provided a useful framework to further develop the more complete and widely applicable platform for PM screening. This study is a contributive one to the clinical management and healthcare work for the myopia patients.

## Methods

**Data collection**. In this study, the use of retinal fundus images was approved by the Ethics Committee of First Affiliated Hospital, School of Medicine, Zhejiang University and adhered to the tenets of the Declaration of Helsinki (Approval ID: NO. 2020–693). Because the study was a retrospective review and analysis of fully anonymized color retinal fundus images, the medical ethics committee declared it exempt from informed consent.

A total dataset was generated, containing 17,330 color retinal fundus images from 13,869 myopia patients aged 14–75 years (Table 1) obtained from the eye center of the First Affiliated Hospital of School of Medicine, Zhejiang University between July 2016 and June 2018. The desktop nonmydriatic retinal cameras and digital retinography systems (Canon) were used to capture the retinal fundus images, which were maculalutea-centered 45° color fundus photographs. The pupil dilation was decided by the examiners depending on the patient's ocular condition. All the data were pseudonymized.

**Definitions and the reference standard**. According to the META-PM study classification, MM was classified into 5 categories: "no myopic retinal degenerative lesion" (Category 0), "tessellated fundus" (Category 1), "diffuse chorioretinal atrophy" (Category 2), "patchy chorioretinal atrophy" (Category 3), and "macular atrophy" (Category 4). Additionally, lacquer cracks, myopic choroidal neovascularization, and Fuchs spot were defined as "plus" lesions[18]. Thus, in this study, eyes

with MM ≥ Category 2 or with at least one of the "plus" lesions were considered as having PM[19], while the low myopic eyes (myopic refractive error between −0.5 D and −6.0 D), and the eyes with MM of Category 0 or Category 1 without "plus" lesions were defined as without PM. The example images of Category 0 to Category 4 MM were shown in Supplementary Fig. 4.

Besides, due to the large volume and high complexity of the raw fundus image data from real-world sources, the quality and gradability of the images were assessed by the ophthalmologists and then used for the further PM grading task. The criteria applied to determine a gradable image was listed below:

1. Image field definition: primary field must include the entire optic nerve head and macula.
2. Images should have perfect exposure because dark and washed-out areas interfere with detailed grading.
3. The focus should be good for grading of small retinal lesions
4. Fewer artifacts: Avoid dust spots, arc defects, and eyelash images
5. There should be no other errors in the fundus photograph, such as the absence of objects in the picture.

After learning the definition and testing the intra- and inter-rater reliability, a total of twenty ophthalmologists from three ophthalmic centers, who achieved an kappa value ≥ 0.81 (almost perfect) for PM and MM category, participated in the manual grading tasks and served as the graders[36]. Fifteen of them were general ophthalmologists with more than five years' experience and five of them were senior retinal specialists with over 10 years' experience. They were randomly grouped into five teams with each team involving one senior specialist. The reference standard of the three tasks was determined based on the following protocol. Graders in the same team evaluated the same set of images. Each grader was blinded to the grading results made by the others and the independent decisions on the fundus images were made. The results recognized unanimously by the three graders in the same team were taken as the reference standard. Results that differed among the general ophthalmologists in the same team were arbitrated by the retinal specialist for final decision of grading[37]. For the detailed workflow of dataset processing, all available fundus images from the total dataset were involved (n = 17330) at the beginning stage, and the gradable/ungradable images were then identified and categorized by the grader teams as described above. Subsequently, in the gradable images group, the PM/non-PM identification and the MM grading (C0 ~ 4 with C0 = non-MM and C1-4 = MM) were completed simultaneously by the grader teams based on the criteria of META-PM classification, with PM or C0-C4 MM images confirmed by the refractive error data (spherical equivalence worse than −6.0 D). The workflow of manual grading tasks was illustrated in Fig. 1. The inclusion/exclusion criteria of three tasks were listed as follows:

1. The binary task: Images obtained from eyes with refractive error (spherical equivalent) worse than −0.5 diopters were included. Images obtained from eyes with previous history of refractive surgery and the ungradable images were excluded.
2. The three-class task: All the images involved in the binary task and the ungradable images excluded in the binary task were all included in the three-class task.
3. The five-class task: Images obtained from eyes with refractive error (spherical equivalent) less than −6.0 diopters, but without serious retinopathy caused by other retinal disease, were included. Images obtained from eyes with previous history of refractive surgery and the ungradable images were excluded.

**Development of the DLSs**. Our training platform was implemented with the PyTorch framework, and all of the DLSs were trained in parallel on four NVIDIA 2080 Ti graphics processing units[38]. After processed by manual grading, the total dataset was randomly subgrouped into the training, validation, and test datasets, which accounted 70, 20, and 10%, respectively. Each image was always taken from a different eye and was only allowed to exist in one dataset, so that no same image would appear in both the training and test datasets. All the raw fundus images were pre-processed by cropping and resizing to meet the requirement of input image format with a resolution of 512 × 512 pixels. The images in the training and validation datasets were pre-processed by the steps of gray-scale transformations, geometric variation and image enhancement, in order to eliminate the irrelevant information and recover the useful or true information in images. The features of input images were then analyzed and extracted by the DLSs. The results of binary and multiclass classifications were given. Meanwhile, the images in the test dataset were simultaneously evaluated by the DLSs and two ophthalmologists (one general ophthalmologist and one retinal specialist external to the aforementioned grading teams). The comparison results between the classifications given by the DLSs and ophthalmologists were used to assess the performance of the DLSs. To further evaluate our DLSs, we also recruited 1000 images from 738 patients aged 17–79 years (Table 1) in another hospital in Shanghai, which were produced by another brand of desktop nonmydriatic retinal camera (NIDEK) and served as an external validation dataset.

In this study, the DLSs were trained using four convolutional neural network (CNN) architectures, namely DenseNet201, ResNet50, VGG16, and Xception. The CNN architectures were evaluated in the test dataset and their performances were

compared. Xception had the best metrics compared with the other architectures in all three tasks (Supplementary Table 3 and Supplementary Fig. 5). Thus, Xception was adopted as the optimal architecture used for the classification tasks. The detailed relevant architecture was shown in Supplementary Fig. 6. Altogether three DLSs were trained to handle binary or multiclass classification tasks: (i) a binary system of none pathologic myopia/pathologic myopia (NPM/PM), (ii) a three-class system of ungradable images/NPM/PM, and (iii) five-class system of five MM categories.

**Misclassification and visualization analysis.** The false-negative and false-positive images misclassified by the three DLSs were further analyzed by a senior retinal specialist. To provide detailed guidance for clinical analysis, a convolutional visualization layer was implanted at the end of the network in the five-class task. Then, this layer generated a visualization heatmap highlighting the strong predicting regions on retinal fundus images[39]. The consistency analysis between the hot regions and the actual lesions was performed by a senior retinal specialist. Additionally, we used an advanced nonlinear dimensionality reduction method, t-distributed stochastic neighbor embedding visualization (t-SNE) to reduce the deep features to two-dimension, and visualize them.

**Statistics and reproducibility.** According to the reference standard in binary task, the performance of the selected general ophthalmologist, retinal specialist, and the DLSs in the test and external validation dataset was calculated using the indices of sensitivity, specificity, accuracy, and area under the receiver operating curve (AUC). For the multi-class tasks, the area under the macro average of ROC (macro-AUC) for each class in one-vs-all manner, quadratic-weighted kappa score, and accuracy were calculated. The source data underlying ROC curves were shown in Supplementary Data 1. We also demonstrated the confusion matrices for the multi-class classification tasks[40]. Additionally, the Clopper–Pearson method was used to calculate the 95% CI. Statistical data were analyzed using Sigma Plot 14.0 and Python 3.7.3.

**Reporting summary.** Further information on research design is available in the Nature Research Reporting Summary linked to this article.

## Data availability

The datasets generated and/or analyzed during the current study are available from the corresponding author on a reasonable request. Correspondence and requests for data materials should be addressed to Wei Han (hanweidr@zju.edu.cn). All datasets are stored at our FTP server. The source data underlying ROC can be found in Supplementary Data 1.

## Code availability

The code used in this study can be accessed at GitHub (https://github.com/18818569575/image_pathological_myopia).

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

## Acknowledgements
This study was supported by grants from the National Natural Science Foundation of China (grant no. 81670842), the Science and technology project of Zhejiang Province (grant no. 2019C03046), the Fundamental Research Funds for the Central Universities (grant no. WK9110000099). The sponsor or funding organization had no role in the design or conduct of this research.

## Author contributions

L.L., E.L.Z., and W.S.Y. had full access to all of the data in the study and take responsibility for the integrity of the data and the accuracy of the data analysis. Concept and design: L.L. and W.H. Administrative, technical, or material support: W.H., L.X.L., D.Q., and L.W. Acquisition, analysis, or interpretation of data: E.L.Z., Q.Y.L., P.F.R., X.Y.T., M.M.Z., Q.Y.L., B.C., and W.S.Y. Drafting of the manuscript: All authors. Critical revision of the manuscript for important intellectual content: All authors. Statistical analysis: E.L.Z. and W.S.Y. Supervision: L.L. and W.H.

## Competing interests

The authors declare no competing interests.
