## [Transparent Peer Review File · Communications Biology]

Reviewers' comments:

Reviewer #1 (Remarks to the Author):

Lu et al present a well-described and excellent designed study to predict pathologic myopia from retinal photographs. The presented work was rigorously carried out, and I appreciate the level of detail in the supplement regarding the deep learning architecture as it is clearly written. However, inclusion criteria should be clearly explained in terms of clinical purpose of this algorithm.

1. Did the authors include only myopic participants or all available retinal photos? If included all available retinal photos then need to start on class of MM vs non-MM. Please clearly explain about target population for clinical application of this deep learning algorithm and inclusion/exclusion criteria.
2. Please provide the information about refractive error for the participants.
3. Please correct the numbers of none PM in Supplementary Table 1
4. Provide the confusion matrix in both internal and external test set.
5. Please provide PR curve in both internal and external test set.
6. Please provide example photos for C0 to C4 and Provide the photos of C3 and C4 photos of internal and external test set in supplementary figure.
7. Number of training set for multiclass task seem to be small, any overfitting?
8. Table 1 should be characteristics of study population including internal and external set.

Reviewer #2 (Remarks to the Author):

The authors propose Deep Learning Systems for 1) Binary classification of non-pathological myopia (NPM) vs pathological myopia (PM), 2) 3-class classification of ungradable vs. NPM vs. PM as well as 3) 5-class myopic maculopathy (MM) stage classification based on retinal fundus images. Using the Xception architecture, authors claim to achieve performance comparable to general ophthalmologists and retinal specialists. Moreover, they collected a dataset of 17330 retinal fundus images labelled by ophthalmologists and senior retinal specialists.

The authors were not the first to apply the CNNs to the classification of PM (<https://doi.org/10.1016/j.procs.2020.01.084>). Also, they are not the first to apply Xception architecture on PM classification (<https://arxiv.org/pdf/2002.06382.pdf>). Both works should be included in the related works. However, the authors are the first to perform a 5-class MM classification.

The reported performance for the proposed DLSs is convincing.

It is not clear, however, why there is no proper comparison with previous works on the binary classification of NPM / PM on the mentioned public databases (e.g. ISBI challenge dataset). A comparison with previous approaches would significantly improve their claims.

Moreover, all results are based on their private internal and external datasets, which cannot be evaluated. Publication of their data would aid reproducibility as well as scientific progress in the community. From the manuscript, it is also not clear whether it was made sure that samples from the same subject do not appear in both training and test set.

An interpretability analysis is included in the form of saliency map computation (based on a variation of the CAM method) and the analysis of misclassified images. Authors claim that the hot regions contained biomarkers relevant for MM lesions. Considering the recent state-of-the-art methods in saliency map computation (e.g. LRP, perturbation-based methods), the chosen method produces weak evidence for clear interpretation due to the coarse resolution and the inability to distinguish between positive and negative contribution. In addition, only 4 hardly recognizable heatmaps are presented in the manuscript and no quantitative measure, constituting poor evidence for the presented claims.

The manuscript could further be improved by including the results of other architectures that are claimed to having been trained as well (e.g. DensNet, ResNet, VGG). Also, confusion matrices should be provided for the external dataset as well and further error analysis is quite important. Examples for the cases where the model had a hard time, reasons, possible solutions. The work would finally benefit from improved phrasing throughout the manuscript.

Reviewer #3 (Remarks to the Author):

This is a straightforward CNN paper on developing and validating a deep learning classification on pathologic myopia.

The strength of the study may include:

1. Large dataset collected from multiple hospitals and cameras.
2. International recognized standard meta-PM was used as the ground truth
3. External validation from different hospital and different camera is available
4. A comparison on the performance between DLS and ophthalmologists with various training background.

Weakness and questions

1. Novelty: adopting deep learning on full colour fundus photo to classify for pathologic myopia is not novel. There have been other publications on similar topic. The study from Kyoko in Japan was on similar topic but they are more detailed in terms of reporting the classification on each category of pathologic myopia and also a classification on lesions.
2. Study population: the authors did not clearly state the composition of the dataset, for example, how many people in each category of the pathologic myopia grading in training set, internal and external validation set.
3. CNN: the authors reported Xception is better than other neuron network and further claimed using Xception is the technologic innovation of the manuscript but they did not provide evidence and explanation on this with sufficient details.
4. Report on performance: the authors described the 3rd task is the classification on each of 5 categories of pathologic myopia but seem to forget report accuracy on this task in the result section when this classification is important for clinical management. For example, a classification on pathologic myopia (diffuse atrophy or worse) is a simple task for any ophthalmologist, but recognizing or classifying a choroidal neovascularization is not that easy but clinically important because they would require anti-VEGF injection.
5. Other minor points: such as Figure 4 heatmap is hard to read, perhaps a high contrast colour would read better., also Figure 2 should have a confusion matrix.

Response letter

Manuscript No. COMMSBIO-20-3516A

**Development of deep learning-based detecting systems for pathologic myopia
using retinal fundus images**

Response to the comments:

We appreciated the editor and reviewers' valuable assessment to help us to improve the manuscript. The appropriate changes were made accordingly in the revised manuscript and printed in red for easy reading.

Referee expertise:

Referee #1: deep learning, ophthalmology, epidemiology

Referee #2: deep learning, biomedical imaging

Referee #3: ophthalmology

Reviewers' comments:

Reviewer #1 (Remarks to the Author):

Comments 1:

Lu et al present a well-described and excellent designed study to predict pathologic myopia from retinal photographs. The presented work was rigorously carried out, and I appreciate the level of detail in the supplement regarding the deep learning architecture as it is clearly written. However, inclusion criteria should be clearly explained in terms of clinical purpose of this algorithm.

Response 1:

We thank the reviewer's support to our work very much.

Comments 2:

However, inclusion criteria should be clearly explained in terms of clinical purpose of this algorithm. Did the authors include only myopic participants or all available retinal photos? If included all available retinal photos then need to start on class of MM vs non-MM. Please clearly explain about target population for clinical application of this deep learning algorithm and inclusion/exclusion criteria.

Response 2:

We included all available retinal photos from the database in the fundus cameras. Therefore, there was no inclusion/exclusion criteria for the original fundus photo database at the beginning screening step of this work. The target population for clinical application of our DLS is the myopic subjects who demand the fundus examination and evaluation for myopic lesions, as the DLS is designed to identify the PM.

Our working flow is: At the beginning step for myopic fundus screening, all original available fundus photos with either PM or non-PM were involved (n=17330). The gradable/ungradable images were firstly identified and categorized by 20 graders. Subsequently, in the gradable images group, the PM/non-PM identification and the MM grading (C0~4 with C0=non-MM and C1-4=MM) were completed based on the criteria of META-PM classification, with PM or C1-C4 MM images confirmed by the refractive error data (spherical equivalence worse than -6.0 D). Hence, the MM and non-MM groups were simultaneously categorized. (See the working flow in Figure 1) The relevant content was added in the revised manuscript. (See in the Method Section, Paragraph 5, Page 14, line 299-306; See in the Discussion Section, Paragraph 2, Page 7-8, line 159-161)

Comments 3: Please provide the information about refractive error for the participants.

Response 3:

Agreed. The refractive error data in each group were provided respectively. (See in the revised Table 1) In view of the poor image quality and discard in the following grading task, the refractive error data of ungradable group were not given.

Comments 4: Please correct the numbers of none PM in Supplementary Table 1

Response 4:

We are sorry about the input error. The typo was corrected accordingly in the revised manuscript. (The Supplementary Table 1 is Supplementary Table 2 in the revised manuscript, See in the Supplementary Table 2)

Comments 5: Provide the confusion matrix in both internal and external test set.

Response 5:

Agreed. The confusion matrix in both internal and external test set was provided in Supplementary Figure 4.

Comments 6: Please provide PR curve in both internal and external test set.

Response 6:

Agreed. The PR curves in both internal and external test set was added in Supplementary Figure 3. The relevant description was added in the revised manuscript. (See in the Result Section, Paragraph 3, Page 6, line 122-123)

Comments 7: Please provide example photos for C0 to C4 and Provide the photos of C3 and C4 photos of internal and external test set in supplementary figure.

Response 7:

Agreed. According to the authoritative literature¹, the example photos for C0 to C4 as well as the photos of C3 and C4 of internal and external test set were provided in Supplementary Figure 5. The relevant description was added in the revised manuscript. (See in the Method Section, Paragraph 3, Page 13, line 281-282)

Comments 8: Number of training set for multiclass task seem to be small, any overfitting?

Response 8:

In the training set of 5-class task, C2-4 images are relatively less. According to a population-based longitudinal study in the Chinese population, only 28% highly myopic eyes had C2-4 MM at baseline. After 10 years of follow-up, only 41% of eyes had C2-4 MM². Several approaches were applied to avoid the overfitting in our work. Firstly, we increased the diversity of the training data by means of data augmentation by randomly flipping training images from left to right or up to down, and adjusting images' brightness, contrast, hue and saturation. Secondly, we inserted dropout layer between the basic Xception model and the last full connection layer. The dropout ration is 0.2 with only 20% of the weights updated every time. Thirdly, we used a two-step training process. In the first training step, only the last full connection layer was fine-tuned and the weights of the basic Xception were not updated. In the second training step, the whole model was fine-tuned and the weights of all the layers were updated. Finally, we used the early stopping strategy. Training was stopped if the validation loss did not decrease in 5 consecutive epochs during the training process. All of these approaches were proved to be able to avoid the overfitting to a large extent. The relevant description was added in the revised manuscript. (See in the Discussion Section,

Paragraph 2, Page 8-9, line 174-184)

Comments 9: Table 1 should be characteristics of study population including internal and external set.

Response 9

We agreed to the reviewer's valuable suggestion. The relevant data regarding characteristics of study population including internal and external dataset were added in the revised manuscript. (See in the revised Table 1)

Reviewer #2 (Remarks to the Author):

The authors propose Deep Learning Systems for 1) Binary classification of non-pathological myopia (NPM) vs pathological myopia (PM), 2) 3-class classification of ungradable vs. NPM vs. PM as well as 3) 5-class myopic maculopathy (MM) stage classification based on retinal fundus images. Using the Xception architecture, authors claim to achieve performance comparable to general ophthalmologists and retinal specialists. Moreover, they collected a dataset of 17330 retinal fundus images labelled by ophthalmologists and senior retinal specialists.

Comments 1: The authors were not the first to apply the CNNs to the classification of PM (<https://doi.org/10.1016/j.procs.2020.01.084>). Also, they are not the first to apply Xception architecture on PM classification (<https://arxiv.org/pdf/2002.06382.pdf>). Both works should be included in the related works. However, the authors are the first to perform a 5-class MM classification.

Response 1

Agreed. The two literatures reported the application of CNN and Xception in PM classification in terms of PM/non-PM or some specific PM lesions. We cited the two articles and refer to their data in the discussion. Changes were made accordingly in the revised

manuscript. (See in the Discussion Section, Paragraph 4, Page 9, line 194-199; See in the Discussion Section, Paragraph 5, Page 10, line 208-209)

Comments 2: The reported performance for the proposed DLSs is convincing. It is not clear, however, why there is no proper comparison with previous works on the binary classification of NPM / PM on the mentioned public databases (e.g. ISBI challenge dataset). A comparison with previous approaches would significantly improve their claims.

Response 2:

The comparison with the previous works, if viable, would be greatly helpful to enhance the plausibility of our work. However, the standard of classification of NPM/PM varied greatly among the multiple public databases (such as ISBI challenge dataset) as well as our dataset. In view of the highly variation of PM classification in the currently available databases, we introduced the META-PM classification, the novel international photographic classification system developed for standard definition and classification of PM, to define PM in our work. This optimized classification unified the previous definitions and is currently widely applied in clinical trials and epidemiologic studies. Therefore, the ground truth for DLS training of our work and the previously reported works are substantially different, it would be unviable to compare the performance between our DLS and the previous works.

Comments 3: Moreover, all results are based on their private internal and external datasets, which cannot be evaluated. Publication of their data would aid reproducibility as well as scientific progress in the community. From the manuscript, it is also not clear whether it was made sure that samples from the same subject do not appear in both training and test set.

Response 3:

We appreciate the reviewer's positive comment on our work. Our work is based on the private datasets from the multiple hospitals. We believe that our work will be contributory to the relevant research area. Our database can be open to the community on request according to the rules of journal Communication Biology. Moreover, it can be sure that each image in the total dataset was taken from the different eye based on the patients'

clinical labels of each image file. Therefore, appearance of images from same participant in both training and test set was definitely avoided. The relevant description was added in the revised manuscript. (See in the Method Section, Paragraph 6, Page14-15, line 312-314)

Comments 4: An interpretability analysis is included in the form of saliency map computation (based on a variation of the CAM method) and the analysis of misclassified images. Authors claim that the hot regions contained biomarkers relevant for MM lesions. Considering the recent state-of-the-art methods in saliency map computation (e.g. LRP, perturbation-based methods), the chosen method produces weak evidence for clear interpretation due to the coarse resolution and the inability to distinguish between positive and negative contribution. In addition, only 4 hardly recognizable heatmaps are presented in the manuscript and no quantitative measure, constituting poor evidence for the presented claims.

Response 4:

Agreed. We redid the Figure 4 heatmaps with higher resolution and higher contrast. In the new heatmaps, it can be easily observed that the red and yellow areas overlap with the characteristic lesions. (See in the revised Figure 4)

In this study, visualization analysis is performed to mainly verify whether the characteristic lesions of each MM category appear in the regions where the algorithm made the positive contribution to the classification results, so that our DLS is justified to be convincing from the clinical point of view. On the other hand, the negative contribution areas are generally not of clinical significance. Referring to some relevant studies in the field of ophthalmology, we chose the CAM method^{3, 4}, which is only focused in the features that have a positive influence on the class of interest and met our requirement very well⁵.

We appreciate the reviewer's valuable suggestion regarding the issues of negative contribution and quantitative measure. In our future work, we will do further in-depth research on the visualization methods.

Comments 5: The manuscript could further be improved by including the results of other architectures that are claimed to having been trained as well (e.g. DensNet, ResNet, VGG).

Also, confusion matrices should be provided for the external dataset as well and further error analysis is quite important. Examples for the cases where the model had a hard time, reasons, possible solutions. The work would finally benefit from improved phrasing throughout the manuscript.

Response 5:

We agree to the reviewer's suggestion. The detailed results of other architectures were reported in the Supplementary Table 1 and Supplementary Figure 1. The relevant explanation was also added in the manuscript. (See in the Method Section, Paragraph 7, Page15, line 329-332)

The confusion matrices of three tasks in the external dataset were provided in Supplementary Figure 4. The distribution of misclassified images for the 3 tasks in the external dataset is essentially the same as that in the test dataset. The relevant description was added in the manuscript. (See in the Result Section, Paragraph 4, Page6-7, line 136-138)

Whilst developing the model, the prevention of overfitting is a critical issue because of the relatively small number of training images for certain categories. Several efforts were made to avoid the overfitting in our work. Firstly, we increased the diversity of the training data by means of data augmentation by randomly flipping training images from left to right or up to down, and adjusting images' brightness, contrast, hue and saturation. Secondly, we inserted dropout layer between the basic Xception model and the last full connection layer. The dropout ration is 0.2 with only 20% of the weights updated every time. Thirdly, we used a two-step training process. In the first training step, only the last full connection layer was fine-tuned and the weights of the basic Xception were not updated. In the second training step, the whole model was fine-tuned and the weights of all the layers were updated. Finally, we used the early stopping strategy. Training was stopped if the validation loss did not decrease in 5 consecutive epochs during the training process. All of these approaches were proved to be able to avoid the overfitting to a large extent. The relevant description was added in the revised manuscript. (See in the Discussion Section, Paragraph 2, Page 8-9, line 174-184)

Reviewer #3 (Remarks to the Author):

This is a straightforward CNN paper on developing and validating a deep learning classification on pathologic myopia.

The strength of the study may include:

1. Large dataset collected from multiple hospitals and cameras.
2. International recognized standard meta-PM was used as the ground truth
3. External validation from different hospital and different camera is available
4. A comparison on the performance between DLS and ophthalmologists with various training background.

Response: We appreciate the reviewer's positive evaluation on our work.

Weakness and questions

Comments 1. Novelty: adopting deep learning on full colour fundus photo to classify for pathologic myopia is not novel. There have been other publications on similar topic. The study from Kyoko in Japan was on similar topic but they are more detailed in terms of reporting the classification on each category of pathologic myopia and also a classification on lesions.

Response 1:

As the work by Kyoko et al. was still not online in the PubMed whilst we submitted this work to the journal, so we did not refer to their work in the original manuscript. We noted that the abstract of article by Kyoko et al. was online in the PubMed in Feb 2021. We congratulate their valuable work on the topic of application of AI in PM screening. In Kyoto's work, they developed four DL algorithms to recognize the Category 2 to 4 MM and CNV. In order to detect PM, these algorithms were also used to develop a META-PM categorizing system (META-PM CS) by adding a specific processing layer. Their definition of PM is having myopic maculopathy equal to or more serious than

diffuse atrophy (Category 2).

Compared with their META-PM CS, our five-class DLS is able to recognize MM from Category 0 to 4 directly. The binary task of NPM/PM was assigned to a well-trained DLS based on a precise PM definition, namely equal to or more serious than diffuse atrophy (category 2) or with at least one of the “plus” lesions.

Besides, our work also has some strength as highlighted in the reviewer’s comments: 1. Large dataset collected from multiple hospitals and cameras. 3, External validation from different hospital and different camera is available. 4. A comparison on the performance between DLS and ophthalmologists with various training background.

In the revised manuscript, we refer to their work in the discussion. (See in the Discussion Section, Paragraph 6, Page10, line 215-221)

Comments 2. Study population: the authors did not clearly state the composition of the dataset, for example, how many people in each category of the pathologic myopia grading in training set, internal and external validation set.

Response 2:

We appreciate the reviewer’s suggestion. We report the detailed information in Supplementary Table 2.

Comments 3. CNN: the authors reported Xception is better than other neuron network and further claimed using Xception is the technologic innovation of the manuscript but they did not provide evidence and explanation on this with sufficient details.

Response 3:

Agreed. The detailed evidence including the confusion matrices and performances of other neuron networks is provided in the Supplementary Table 1 and Supplementary Figure 1. The relevant explanation is also added in the manuscript. (See in the Method Section, Paragraph 7, Page 15, line 329-332)

Comments 4. Report on performance: the authors described the 3rd task is the classification on each of 5 categories of pathologic myopia but seem to forget report

accuracy on this task in the result section when this classification is important for clinical management. For example, a classification on pathologic myopia (diffuse atrophy or worse) is a simple task for any ophthalmologist, but recognizing or classifying a choroidal neovascularization is not that easy but clinically important because they would require anti-VEGF injection.

Response 4:

We agree to the reviewer's valuable suggestion. The appropriate data were added accordingly in the revised manuscript and Table 3. (See in the Result Section, Paragraph 3, Page 6, line 118-120)

Comments 5. Other minor points: such as Figure 4 heatmap is hard to read, perhaps a high contrast colour would read better, also Figure 2 should have a confusion matrix.

Response 5:

We agreed with the reviewer regarding the Figure 4 heatmap. We redid the Figure 4 heatmap with higher resolution and higher contrast colour. In the new heatmaps, it can be easily observed that the red and yellow areas overlap with the characteristic lesions. (See in the revised Figure 4) The corresponding confusion matrixes were also provided in the Supplementary Figure 4a-c.

End of response.

References:

- 1 Ohno-Matsui K, Kawasaki R, Jonas JB, et al. International Photographic Classification and Grading System for Myopic Maculopathy. *Am J Ophthalmol.* 2015;159(5):877-883.
- 2 Yan YN, Wang YX, Yang Y, et al. Ten-Year Progression of Myopic Maculopathy. *Ophthalmology.* 2018;125(8):1253-1263.
- 3 Gargeya R, Leng T. Automated Identification of Diabetic Retinopathy Using Deep Learning. *Ophthalmology.* 2017;124(7):962-969.
- 4 Liu H, Li L, Wormstone IM, et al. Development and Validation of a Deep Learning System to Detect Glaucomatous Optic Neuropathy Using Fundus Photographs. *Jama Ophthalmol.* 2019.
- 5 Rs R, Cogswell M, Das A, et al. Grad-CAM: Visual Explanations from Deep Networks via Gradient-Based Localization. *Int J Comput Vision.* 2020;128.

Reviewers' comments:

Reviewer #1 (Remarks to the Author):

The reviewer recommends inviting one of the META-PM study group members to confirm whether the use of META-PM classification in non-high myopic participants is appropriate or not.

The reviewer asked about inclusion criteria in comment 2 because the META-PM classification was developed based on patients with only high myopia at the High Myopia Clinic of the Tokyo Medical and Dental University. (Please correct me if I am wrong.).

In the present study, the authors included "non-high myopic (not non-myopic)" participants in C0 classification, which inflates the accuracy due to the original C0 class defined as "no myopic retinal changes" among "only" high myopic participants.

This is also directly related to the purpose of this study. As the authors mention in line 159-161, if the authors intended to detect PM among the subjects with myopia, then need to exclude META-PM results or need to present separately using only high myope. In addition, kindly define pathologic myopia and the reviewer believes every study has its own inclusion/exclusion criteria according to the purpose of the study.

Reviewer #5 (Remarks to the Author):

This manuscript proposed a deep learning application on pathologic myopia image classification. This work collects a large set of PM image data and applies appropriate strategies to train/validate/test the model. The results look promising, for most of the images indeed have explicit differences between categories, the proposed model should get these excellent results as expected. The response letter addressed most of the comments from the former reviewers, and the revision significantly enhanced the manuscript.

From my perspective, there is nothing wrong with this work, it is definitely publishable somewhere. The flaws that may impede its impact are:

1. The accessibility of the image data as reviewer 2 mentioned. It is common that authors hold these data for many reasons, but it causes difficulties for reproducibility. Of course, we should respect the authors' choice here.
2. Even open-sourced the codes in Github, the software lacks some user-friendly features as a tutorial for other users who are interested in this work.
3. It will be excellent if this work can shed some light on biology interpretation, such as checking low-level learned features in CNN layers, and how these learned features are related to PM in the study.
4. It will be great if this work can discuss why Xception outperforms other models from a methodology perspective, which module/trick Xception captures better than other models, how these tricks are linked with the PM mechanisms on the images.
5. For these models trained for 3 related classification tasks, can we find some inner links (shared weights or something) on these models? If we check the confidence of the prediction in these tasks, can we find some patterns in different stages in binary classification?
6. Sometimes the data quality is so good that most methodologies perform great. If we can have a PCA/UMAP/t-SNE plot on all the data used, it may already have a good pattern visually between these images as categories.

Minor:

Line 91: 16428 should be 16,428

Response letter

Manuscript No. COMMSBIO-20-3516B

**Development of deep learning-based detecting systems for pathologic myopia
using retinal fundus images**

Response to the comments:

We appreciated the editor and reviewers' valuable assessment to help us to improve the manuscript. The appropriate changes were made accordingly in the revised manuscript and printed in red for easy reading.

Reviewers' comments:

Reviewer #1 (Remarks to the Author):

Comments 1:

The reviewer recommends inviting one of the META-PM study group members to confirm whether the use of META-PM classification in non-high myopic participants is appropriate or not.

The reviewer asked about inclusion criteria in comment 2 because the META-PM classification was developed based on patients with only high myopia at the High Myopia Clinic of the Tokyo Medical and Dental University. (Please correct me if I am wrong.).

In the present study, the authors included “non-high myopic (not non-myopic)” participants in C0 classification, which inflates the accuracy due to the original C0 class defined as "no myopic retinal changes” among “only" high myopic participants.

This is also directly related to the purpose of this study. As the authors mention in line 159-161, if the authors intended to detect PM among the subjects with myopia, then need to exclude META-PM results or need to present separately using only high myope. In addition, kindly define pathologic myopia and the reviewer believes every study has its own inclusion/exclusion criteria according to the purpose of the study.

Response 1:

We would like to thank the reviewer for the pertinent suggestion. After consulting with the META-PM study group member and referring to the relevant literatures, we confirmed again that all images in the C0 classification should have been from patients with high myopia when the META-PM classification was developed. We are very grateful for the reviewer's critical comments and sorry about our negligence when involving the C0 participants.

Hence, in the five-class task, we excluded all C0 images from non-highly myopic participants and re-tested the DLS only in high myopia. According to the results, we found a mild decrease in accuracy (0.987 to 0.976), with almost no change in AUC and kappa value. (See in the Result Section, Paragraph 3, Page 6, line 117-118, Paragraph 4, Page 6, line 132-135; See in the Method Section, Paragraph 5, Page 14, line 309-310; See in the revised Figure 1, Figure 3, Table 1, Table 3, Supplementary Table 1, Supplementary Table 2, Supplementary Table 3, Supplementary Figure 1, Supplementary Figure 3, Supplementary Figure 4, Supplementary Figure 5)

On the other hand, after checking with the relevant experts and reviewing the relevant literatures, we believed that the META-PM classification and the definition of PM based on the META-PM classification can be applied to screening studies involving low grade myopia and emmetropic eyes, like the prevalence investigation of PM¹⁻³. Of course, the primary function of our system is to identify the PM lesions in high myopia individuals. Together, the binary task and three-class task here were basically not affected by the C0 participants change. In the present work, the PM definition based on the META-PM classification can also be applied. Moreover, the inclusion/exclusion criteria for each of the three tasks were also added in the supplementary information. (See in the Method Section, Paragraph 5, Page 14, line 310-311; Supplementary Note2)

Reviewer #5 (Remarks to the Author):

This manuscript proposed a deep learning application on pathologic myopia image classification. This work collects a large set of PM image data and applies appropriate strategies to train/validate/test the model. The results look promising, for most of the images indeed have explicit differences between categories, the proposed model should get these excellent results as expected. The response letter addressed most of the comments from the former reviewers, and the revision significantly enhanced the manuscript.

From my perspective, there is nothing wrong with this work, it is definitely publishable somewhere. The flaws that may impede its impact are:

Comments 1:

The accessibility of the image data as reviewer 2 mentioned. It is common that authors hold these data for many reasons, but it causes difficulties for reproducibility. Of course, we should respect the authors' choice here.

Response 1:

We thank the reviewer for understanding the difficulties of the work. Building a fully publicly available dataset needs to complete multiple procedures to meet relevant policies and regulations. Our team is continuously putting effort in this area and hopefully be able to release a public dataset of pathologic myopia and myopic maculopathy in the future. For time being, the images involved in the study are available kindly by contacting the corresponding author.

Comments 2:

Even open-sourced the codes in Github, the software lacks some user-friendly features as a tutorial for other users who are interested in this work.

Response 2:

We agree to the reviewer's valuable suggestion. Some engineering attempts have been made using the DLSs reported in the manuscript and the relevant software has been successfully developed. Readers can contact the corresponding author for a beta version of the software.

Comments 3:

It will be excellent if this work can shed some light on biology interpretation, such as checking low-level learned features in CNN layers, and how these learned features are related to PM in the study.

Response 3:

Agreed. We have tried to visualize the low-level learned features in CNN layers and found that the contours, edges, and shape features of the pixel points in the lesion

regions contrasted significantly with normal areas. This result demonstrates that the underlying image features learned by the CNN are more perceptive of the lesion regions and can effectively capture the semantic relatedness of the lesion and normal regions, indicating the connection between the underlying image features learned by the CNN and the characteristic lesions of the PM. (Figure 1)

Comments 4:

It will be great if this work can discuss why Xception outperforms other models from a methodology perspective, which module/trick Xception captures better than other models, how these tricks are linked with the PM mechanisms on the images.

Response 4:

We appreciate the reviewer's suggestion. In Xception architecture, Inception modules were replaced with depthwise separable convolutions to handle spatial-correlations and cross-channel correlations independently. It is thought that an independent processing of cross-channel correlations would better address the tasks involved in this study. Due to the strong pixel similarity in the characteristic PM lesions, Xception can better learn the characteristic similarity of multiple pixel points within a PM lesion to make more accurate judgments by processing cross-channel correlations independently. The relevant description was added in the revised manuscript. (See in the Discussion Section, Paragraph 2, Page 8, line 172-177)

Comments 5:

For these models trained for 3 related classification tasks, can we find some inner links (shared weights or something) on these models? If we check the confidence of the prediction in these tasks, can we find some patterns in different stages in binary classification?

Response 5:

We thank the reviewer for this constructive suggestion. The three related classification tasks were all trained on the same total dataset. The images in the dataset were classified differently to correspond to each task, depending on the

purpose. Therefore, instead of developing the rest of the models based on the model of one task, we trained the three models independently.

According to the reviewer's suggestion, we tried to use the weights of the 2-classification model as the initial weights to train the 3-classification and 5-classification models. After comparison, the training weights of the 2-classification model can effectively serve as the initialisation weights for the 3 and 5 classification models, improving the accuracy by approximately 9% and 11% respectively over the original models at the very beginning of the training phase. This indicates that the model of the 2 classification task can be used as a pre-training model for the 3 and 5 classification tasks, thus demonstrating that there are some inner links such as data distribution and feature representation between the 3 tasks, but the performances of the new 3 and 5 classification models at the final phase are not significantly different from that of the original independently trained models. (Figure 2)

In the subsequent studies of other fundus diseases from different datasets, we will try to develop new models based on the model in present study, in order to further investigate the inner links between the models.

We examined the confidence of the prediction in these tasks, and noted that the majority of images in each task had significant variance between classification prediction scores. However, there were still a few images with poor confidence. By examining the literatures, we believe that the following patterns can be tried in the training and testing phases.

1. We use a training strategy based on MC Dropout. This method achieves the measure of uncertainty by randomly sampling the model parameters K times. In the training phase, the model outputs K classification prediction results, and then the model's prediction and confidence are obtained by calculating the mean and variance of the results. In the test phase, the dropout is not turned off, K dropouts are performed, K forward passes are made, K results are output, and then ensemble is performed to give the model's prediction and uncertainty⁴.

2. We perform multiple random sampling from the same dataset to train the models separately, and then combine the results of these models, with the mean as the prediction result and the variance as the prediction uncertainty.

These patterns provide adjustment and optimization for the confidence of the prediction.

Comments 6:

Sometimes the data quality is so good that most methodologies perform great. If we can have a PCA/UMAP/t-SNE plot on all the data used, it may already have a good pattern visually between these images as categories.

Response 6:

We agree to the reviewer's suggestion. The t-SNE plots of three tasks were added in Supplementary Figure 6. The relevant descriptions were also included in the manuscript. (See in the Result Section, Paragraph 5, Page 7, line 146-149; Supplementary Figure 6)

Minor:

Line 91: 16428 should be 16,428

Response:

The typo was corrected accordingly in the revised manuscript. (See in the Result Section, Paragraph 1, Page 5, line 99)

End of response.

Figure 1: Visualization of the low-level learned features in CNN layers. (a) The original images of PM (Category4). (b) Low-level feature map visualization

Figure 2: The comparison of the original models and new models. (a) The 3-classification models. (b) The 5-classification models

References

1. Koh, V. et al. Myopic Maculopathy and Optic Disc Changes in Highly Myopic Young Asian Eyes and Impact on Visual Acuity. *Am. J. Ophthalmol.* **164**, 69-79 (2016).
2. Hopf, S. et al. Prevalence of myopic maculopathy in the German population: results from the Gutenberg health study. *Brit. J. Ophthalmol.*, 2019-315255 (2019).
3. Ohno-Matsui, K. WHAT IS THE FUNDAMENTAL NATURE OF PATHOLOGIC MYOPIA? *Retina.* **37**, 1043-1048 (2017).

4. Gal, Y. & Ghahramani, Z. Dropout as a Bayesian Approximation: Representing Model Uncertainty in Deep Learning. *Proceedings of The 33rd International Conference on Machine Learning*, (2015).